# RNA Sequencing of Pooled Samples Effectively Identifies Differentially Expressed Genes

**DOI:** 10.3390/biology12060812

**Published:** 2023-06-02

**Authors:** Bokang Ko, Jeremy M. Van Raamsdonk

**Affiliations:** 1Department of Neurology and Neurosurgery, McGill University, Montreal, QC H3A 2B4, Canada; bokang.ko@mail.mcgill.ca; 2Metabolic Disorders and Complications Program (MeDiC), Research Institute of the McGill University Health Centre, Montreal, QC H4A 3J1, Canada; 3Brain Repair and Integrative Neuroscience Program (BRaIN), Research Institute of the McGill University Health Centre, Montreal, QC H4A 3J1, Canada; 4Division of Experimental Medicine, Department of Medicine, McGill University, Montreal, QC H4A 3J1, Canada

**Keywords:** RNA sequencing, gene expression, microarray, reproducibility, concordance, differentially expressed genes

## Abstract

**Simple Summary:**

Gene expression studies provide valuable insights into the mechanisms underlying biological processes, including aging. RNA sequencing can be used to identify changes in gene expression across the entire genome. While normally, RNA sequencing is performed on multiple biological replicates that are sequenced individually, the results of this study show that similar results can be obtained by pooling those individual samples together before sequencing. Pooling RNA samples prior to sequencing will reduce the cost of experiments, which may allow for additional investigations and provide more information about the mechanisms involved.

**Abstract:**

Analysis of gene expression changes across the genome provides a powerful, unbiased tool for gaining insight into molecular mechanisms. We have effectively used RNA sequencing to identify differentially expressed genes in long-lived genetic mutants in *C. elegans* to advance our understanding of the genetic pathways that control longevity. Although RNA sequencing costs have come down, cost remains a barrier to examining multiple strains and time points with a sufficient number of biological replicates. To circumvent this, we have examined the efficacy of identifying differentially expressed genes by sequencing a pooled RNA sample from long-lived *isp-1* mitochondrial mutant worms. We found that sequencing a pooled RNA sample could effectively identify genes that were found to be significantly upregulated in the two individually sequenced RNA-seq experiments. Finally, we compared the genes significantly upregulated in the two individually sequenced RNA-seq experiments to two previous microarray experiments to come up with a high-confidence list of modulated genes in long-lived *isp-1* mutant worms. Overall, this work demonstrates that RNA sequencing of pooled RNA samples can be used to identify differentially expressed genes.

## 1. Introduction

Aging is the greatest risk factor for the development of many neurodegenerative diseases, including Alzheimer’s and Parkinson’s disease [1,2,3,4]. Interestingly, genes and interventions that extend lifespan in model organisms have been shown to be neuroprotective in animal models of neurodegenerative disease [5,6,7,8,9,10,11,12,13,14]. This suggests that targeting aging pathways may be beneficial in neurodegenerative disease and that by advancing our knowledge of the aging process, the insights gained might be applied to develop novel treatments for these devastating disorders.

In order to increase our understanding of the aging process, we and others have looked for changes in gene expression in long-lived genetic mutants to better understand the molecular mechanisms that contribute to lifespan extension. For example, we have used RNA sequencing (RNA-seq) to identify gene expression changes in long-lived mitochondrial mutants, including *clk-1* [15,16], *isp-1* [17], and *nuo-6* [18]. These studies found that genetic targets from multiple pathways of cellular resilience are upregulated in the long-lived mitochondrial mutants and required for their extended longevity [19,20,21,22,23]. These gene expression studies also demonstrated a high degree of overlap between genes correlated with resistance to stress and genes correlated with longevity, suggesting that the same genetic pathways contribute to both phenotypes [24]. In addition to our work, a number of other laboratories have used either RNA-seq or microarrays to identify gene expression changes in long-lived genetic mutants [25,26,27,28,29,30,31,32].

Gene expression analysis has been widely used to investigate how cells differentially regulate the expression of certain gene products in the host organism. The most commonly used techniques to detect significant changes in transcript levels include quantitative reverse transcriptase PCR (qRT-PCR) [33], microarrays [34,35], and RNA-seq [36,37,38].

qRT-PCR is a cost-effective and accessible method to analyze gene expression in which even minute quantities of mRNA can be analyzed [39]. After isolating RNA, mRNA is reverse-transcribed into cDNA. Using primers designed to specifically bind to the cDNA sequence, and not genomic DNA, the gene or target of interest is then amplified in the presence of a fluorescent reporter molecule such as SYBR Green, which binds to DNA, such that the increase in DNA generated by the PCR reaction can be monitored in a specialized thermal cycler that can detect fluorescence. While qRT-PCR is easy to perform and cheaper than other gene expression approaches, only known sequences can be targeted for analysis, hence limiting its use in explorative studies. qRT-PCR is also only feasible to examine the expression of a limited number of genes and is best suited for quantifying specific genetic targets.

In contrast to qRT-PCR, microarray technology allows for the quantification of all or most of the genes in the genome. In this approach, many thousands of nucleic acid sequences representing all or part of the genome are bound to a chip [40]. mRNA from the sample of interest is then labelled with a fluorescent marker and hybridized to the chip. In this way, the mRNA can be detected by fluorescence scanning, where the intensity of the fluorescence signal is directly proportional to the amount of transcript present. While microarrays provide a relatively unbiased, exploratory approach to identify the genetic underpinnings of the biological process of interest, they are still reliant upon existing transcript sequences as with qRT-PCR. Despite high sample throughput, microarrays display high background signals and cross-hybridization, which reduces their sensitivity [41].

Unlike qRT-PCR or microarray, RNA-seq is capable of both targeted and unbiased analysis of gene expression, allowing for the identification and quantification of novel and known transcripts. RNA-seq works by sequencing extracted and purified RNA samples, and the sequence reads are either counted to quantify expression levels or assembled de novo for a genome-scale transcript mapping [42,43]. Not only does RNA-seq allow the quantification of gene products, but it also detects other transcriptomic information, such as splice variants and isoforms, making it a powerful tool for gene expression analysis [43]. However, the cost of performing RNA-seq limits its widespread use.

To circumvent the financial limitation of using RNA-seq, we examined the efficacy of sequencing pooled RNA samples from individual biological replicates to identify differentially expressed genes. To ensure that pooling does not affect RNA-seq results, we compared differentially expressed genes in wild-type and *isp-1* worms by sequencing both pooled and individual mRNA samples. We found no difference in the ability to identify differentially expressed genes when individual RNA samples are pooled together into a single sample, thereby lowering the experimental cost of performing RNA-seq and making sequencing more accessible.

## 2. Materials and Methods

### 2.1. Strains

Wild-type N2 worms and *isp-1(qm150)* worms were used in this study. N2 worms were isolated from the wild in Bristol, England, and are the most commonly used wild-type strain for *C. elegans* research. *isp-1* worms were isolated in an ethyl methane sulfonate (EMS) mutagenesis screen for slow development [17]. Worms were grown on nematode growth medium (NGM) plates and 20 °C and fed OP50 bacteria.

### 2.2. Isolation of mRNA

Pre-fertile young adult worms from a 24 h limited lay were collected and washed three times in M9 buffer. After pelleting the worms and removing M9 buffer, worms were frozen in Trizol (ThermoFisher Scientific, Waltham, MA, USA) in liquid nitrogen and stored at −80 °C until RNA extraction. RNA for each experiment was all isolated at the same time as we have carried out previously [44]. To isolate RNA, collected worms underwent three freeze–thaw cycles, three cycles of 30 s of vortexing followed by 30 s on ice, and then were left to sit for 15 min at room temperature. Chloroform was added at 1:5 volume of Trizol and then vortexed for 15 s and allowed to sit at room temperature for 3 min. Samples were centrifuged at 12,000× *g* for 15 min at 4 °C, and then the upper aqueous phase was transferred to a new tube and mixed with isopropanol at 1:2 volume Trizol. After sitting at room temperature for 10 min, samples were centrifuged at 12,000× *g* for 10 min at 4 °C. The supernatant was removed, and the pellet was washed with 75% ethanol before centrifuging for 10 min at 4 °C. After removing ethanol and briefly air drying the pellet, it was resuspended in RNase free double distilled water. RNA samples were frozen and stored in a −80 °C freezer. RNA quality and concentration were initially determined using a Nanodrop spectrophotometer and subsequently measured using an Agilent Bioanalyzer prior to library preparation.

### 2.3. RNA Sequencing and Determination of Differentially Expressed Genes

Three different RNA sequencing paradigms were used to identify differentially expressed genes in *isp-1* worms compared to wild-type worms. For each biological replicate, one 60 mm plate containing hundreds to thousands of worms from a 24 h limited lay was collected on a different day. In one experiment, RNA was isolated individually from six biological replicates of wild-type and *isp-1* worms and sequenced individually [23]. In a second experiment, RNA was isolated individually from nine biological replicates of wild-type and *isp-1* worms and then pooled into three samples prior to sequencing [19]. In a third experiment, RNA was isolated independently from six biological replicates of wild-type and *isp-1* worms. The wild-type RNA samples were sequenced individually, while the *isp-1* samples were pooled in equal amounts to form one sample prior to sequencing. For all three experiments, sequencing was performed on an Illumina NextSeq500 sequencer. Quality control, analysis of the sequencing results, and determination of differentially expressed genes were performed in an identical manner for all three experiments as described previously [19]. The sequencing data from these experiments have been deposited at NCBI GEO: GSE95240 and GSE93724. An overview of the sample collection and sequencing can be found in Appendix A.

### 2.4. Criteria to Identify Differentially Expressed Genes

A number of different criteria were used to identify differentially expressed genes. For percentage increase (500%, 200%, 150%), the average gene expression level in the *isp-1* samples was divided by the average gene expression level in the wild-type samples. If the result was above the threshold of 500%, 200%, or 150% increase, the gene was designated a differentially expressed gene. For standard deviation increase (3 STDEV, 2 STDEV), the standard deviation was calculated for the expression levels in the individually sequenced wild-type samples. The difference in gene expression level between the average of the *isp-1* samples and the average of the wild-type samples was divided by the standard deviation to express the difference in terms of number of standard deviations. If this number was above the threshold of 3 STDEV or 2 STDEV increase, then the gene was designated a differentially expressed gene. For the criteria that combined both percent increase and standard deviation, a gene needed to satisfy both criteria (e.g., percentage increase of 150% and an increase of 2 STDEV) to be designated a differentially expressed gene.

### 2.5. Comparison of Differentially Expressed Genes

Venn diagrams comparing differentially expressed genes between different experiments were generated using BioVenn (www.biovenn.nl, accessed on 13 May 2023) [45]. To facilitate comparison between different datasets, all differentially expressed genes were identified using their WormBase ID using Wormmine (http://intermine.wormbase.org/tools/wormmine/begin.do, accessed on 13 May 2023) [46]. In some cases, genes that were previously reported to be differentially expressed are no longer considered to be genes (e.g., if the putative gene was found to be of transposon origin).

## 3. Results

### 3.1. Reproducibility of RNA Sequencing Results between Experiments

In order to determine the reproducibility of RNA-seq results between experiments, we compared genes that were found to be significantly upregulated in long-lived *isp-1* mutants from two separate studies. In both studies, the *isp-1* worms were collected at the pre-fertile young adult stage, and the collection of worms, preparation of mRNA, RNA sequencing, and data analysis were performed in the same way. In the first study, nine biological replicates, each consisting of a plate with hundreds to thousands of individual worms, were pooled into three samples prior to sequencing, while in the other experiment, six biological replicates, each consisting of a plate with hundreds to thousands of individual worms, were sequenced individually (Figure 1A).

In comparing genes that were significantly upregulated in *isp-1* worms compared to wild-type worms with a false discovery rate (FDR) less than 0.05, we found that there was a 52% overlap (Figure 1B, Appendix A). While the overlap of 735 genes is highly significant (fold enrichment = 6.9; *p* = 0), in both gene sets, there were also many significantly upregulated genes that were not identified by the other experiment (685 and 776 genes).

To determine if the genes that did not overlap were modulated in a similar manner but failed to reach significance, we compared these gene sets using a number of different criteria that were not based on significance. We compared genes that were upregulated by 500%, 200%, 150%, 3 standard deviations, 2 standard deviations, 150% and 3 standard deviations or 150% and 2 standard deviations. Similar to comparing genes that were significantly upregulated, we found that there was a high degree of overlap ranging from 49% for genes upregulated by 200% to 70% for genes upregulated by 2 STDEV (Appendix A), but that there were still a number of genes that were only modulated in one gene set or the other.

### 3.2. Differentially Expressed Genes Can Be Identified by Sequencing a Single Pooled RNA Sample

Based on our comparison between two different RNA-seq experiments, we wondered whether it would be possible to identify differentially expressed genes by sequencing a pooled RNA sample. To this end, we collected six biological replicates of wild-type and *isp-1* worms and isolated mRNA individually from each sample. As with the individually sequenced RNA samples, each replicate/sample contained hundreds to thousands of individual worms from the same plate. For the *isp-1* sample, we then combined equal amounts of each mRNA isolated from the six samples into one pooled sample (Figure 1A). We then sequenced the pooled *isp-1* sample and identified genes that are upregulated compared to wild-type using the same criteria as we used for the *isp-1* mRNA samples that were sequenced individually. These upregulated genes were then compared to upregulated genes identified in the two experiments in which *isp-1* samples were sequenced individually. In comparing genes upregulated in *isp-1* worms that were identified by different criteria, the percentage overlap between the three RNA-seq experiments ranged from 47% when comparing genes increased by 200% to 63% for genes increased by two standard deviations (Figure 1).

To determine the efficacy of identifying differentially expressed genes from the pooled RNA-seq experiment, we compared genes found to be upregulated in the pooled RNA-seq sample to genes found to be significantly upregulated in both of the two individually sequenced RNA experiments. We found that analysis of the pooled RNA-seq experiment could identify 15–95% of the 735 genes that were found to be significantly upregulated in both individually sequenced RNA-seq experiments with an accuracy ranging from 21–35% (Figure 2, Appendix A). The largest number of genes was identified by finding genes that were increased by two standard deviations (701 genes with 21% accuracy). The greatest accuracy was achieved by finding genes that were increased by 500% (111 genes with 35% accuracy). Overall, multiple criteria were effective in identifying significantly upregulated genes from the pooled RNA-seq experiment.

To further examine the efficacy of identifying modulated genes by sequencing a pooled RNA sample, we examined the overlap between pairwise comparisons using the criteria outlined above. We examined both the number of overlapping genes (Figure 3A and Appendix A) and the percentage overlap (Figure 3B). Compared to the 735 genes that were identified by comparing upregulated genes with FDR < 0.05, the criteria of genes upregulated by 150%, genes upregulated by 3 standard deviations, and genes upregulated by 2 standard deviations identified a similar number or more overlapping genes (Figure 3A). For every criterion, a greater number of overlapping genes were found for comparisons between the pooled RNA-seq experiment and either of the individually sequenced RNA-seq experiments than for the comparison between the two individually sequenced RNA-seq experiments. The highest number of overlapping upregulated genes was achieved by comparing genes upregulated by 2 standard deviations. Compared to the 52% overlap achieved by comparing the upregulated genes with FDR < 0.05, all of the other criteria achieved a similar or greater percentage overlap (Figure 3B). In every comparison, the gene set obtained by individually sequencing six biological replicates exhibited the greatest degree of overlap with the other two gene sets and the fewest non-overlapping genes. Combined, these results suggest that pooling mRNA samples prior to RNA sequencing does not reduce the ability to identify differentially expressed genes.

### 3.3. Overlap between Independent Microarray Experiments and RNA Sequencing

Two previous studies have examined gene expression in *isp-1* worms using microarray technology. Cristina et al. used Illumina microarrays to examine the L4 stage (2 biological replicates) and pre-fertile adults (4 biological replicates), which were combined for analysis [25]. Yee et al. examined young adult worms on Affymetrix *C. elegans* GeneChips [26]. We compared the results of these two previous studies to our RNA-seq results. For this purpose, we used genes that were found to be significantly upregulated or downregulated in both of our RNA-seq datasets that were sequenced individually (FDR < 0.05). We found that there were very few genes found to be significantly upregulated (54 genes; Figure 4A, Appendix A) or significantly downregulated (18 genes; Figure 4B, Appendix A) in all of the gene sets. There was a 53% overlap of both upregulated and downregulated genes between our combined RNA-seq results and the microarray results from Yee et al. [26]. In contrast, there was only a 13% overlap in upregulated genes and a 25% overlap in downregulated genes between our combined RNA-seq results and the microarray results from Cristina et al. [25].

## 4. Discussion

Identification of differentially expressed genes using genome-wide gene expression studies provides an unbiased approach to gain insight into how genes and interventions affect biological processes. The two main approaches used in these studies have been microarray and RNA-seq [47]. Previous studies have observed a significant degree of overlap in comparing microarray results to RNA-seq results [48,49,50,51,52]. In general, a higher percentage of genes identified by RNA-sequencing is confirmed by qRT-PCR compared to results from microarray [48,53]. Interestingly, some researchers have developed methods to combine RNA-seq and microarray data together [54].

In this work, we compared the differentially expressed genes identified in three different RNA-seq experiments comparing long-lived *isp-1* mutants to wild-type worms. Our results indicate that there is a high degree of overlap between the results of different RNA-seq studies that are independent of whether the RNA samples were sequenced individually or pooled and sequenced together. The degree of overlap between differentially expressed genes identified by two experiments in which the RNA samples were sequenced individually was similar to the degree of overlap with genes identified by sequencing RNA samples together as a pooled sample. This suggests that sequencing pooled RNA samples can be as effective at identifying differentially expressed genes as sequencing individual samples. By using pooled RNA samples, the cost of RNA-seq will be reduced, which would allow for sequencing more genotypes, conditions, or time points.

In our experiments, we compared pooled *isp-1* samples to wild-type samples that were sequenced individually. This was carried out so that we could compare the efficacy of using percentage change to the efficacy of using standard deviation to identify differentially expressed genes. Our results suggest that using percentage change can be comparably effective to using standard deviation as a criterion to identify differentially expressed genes. Based on this, it should be possible to sequence pooled samples for the wild-type control worms as well.

Potential limitations of sequencing a pooled RNA sample include losing information about the variability in gene expression levels and the inability to determine whether the differences observed are statistically significant. However, these two limitations can be overcome by confirming the result for genes of interest using qRT-PCR, as qRT-PCR will provide information about the levels of gene expression between individual samples and allow for statistical comparisons. In order to confirm differences using qRT-PCR, only a portion of the RNA sample should be pooled for sequencing and a portion saved for qRT-PCR.

Another important consideration is which organism is being examined. In genetic model organisms such as *C. elegans*, *Drosophila*, or inbred mice, each animal has a genetically identical background. In contrast, in studies involving humans or animals from the wild, there is genetic variation across the whole genome. In the latter case, it may be important to sequence samples individually to not lose information about variability that is caused by differences between individual genomes.

In addition to comparing differentially expressed genes from different RNA-seq experiments, we also compared our results to previous microarray studies. While we observed a similar degree of overlap between the genes identified by Yee et al. [26] and genes identified in our RNA-seq studies, much less overlap was observed with genes identified by Cristina et al. [25]. It is uncertain whether this is due to differences in microarray technology (Affymetrix versus Illumina), the fact that Cristina et al. [25]. combined results for adults and larval worms, or other experimental differences. Nonetheless, we did identify 54 genes that were significantly upregulated in all three datasets and 18 genes that were significantly downregulated in all of the datasets. These genes represent high-confidence differentially expressed genes in *isp-1* mutants that may provide insights into the molecular mechanisms underlying their lifespan extension.

## 5. Conclusions

Overall, our results indicate that RNA sequencing can be used as a reproducible approach to identify differentially expressed genes in order to gain insight into the molecular mechanisms underlying the aging process. Pooling individually isolated RNA samples prior to sequencing can be as effective at identifying differentially expressed genes as sequencing RNA samples individually, especially when results will be confirmed with quantitative PCR. Sequencing pooled RNA samples will reduce experimental costs or allow for more time points or genotypes or interventions to be tested.

## Figures and Tables

**Figure 1 biology-12-00812-f001:**
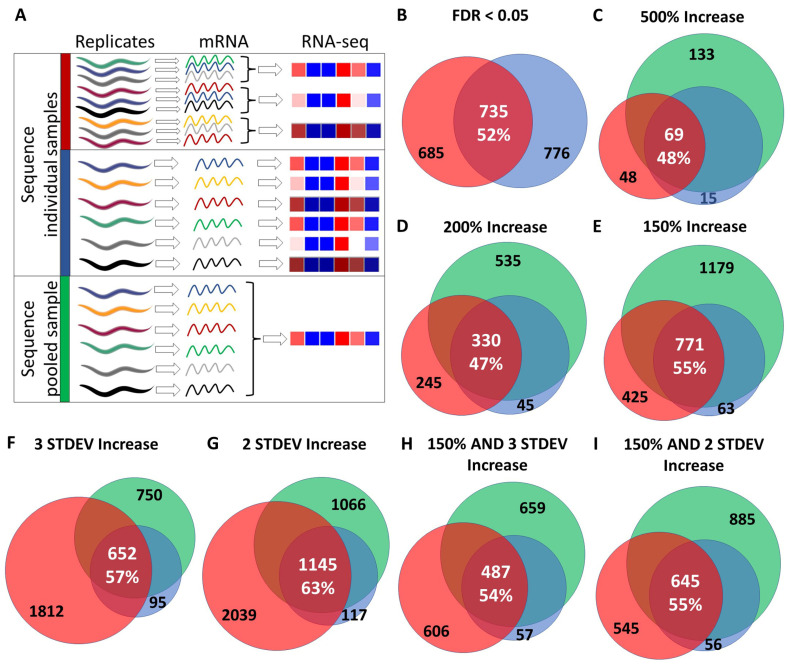
Reproducibility of differentially expressed genes identified by RNA sequencing. (**A**) To assess the reproducibility of RNA sequencing across different experiments, we compared the differentially expressed genes (DEGs) in long-lived *isp-1* mutants to wild-type worms in three separate experiments. In two experiments, either three (red) or six (blue) biological replicates were collected and sequenced individually. To assess the efficacy of using pooled RNA sequencing to identify differentially expressed genes (DEGs), we also compared the results to a third experiment in which RNA was isolated from six individual biological replicates and then pooled prior to sequencing (green). In the diagram, the worm image represents a sample containing hundreds to thousands of individual worms. We examined the degree of overlap from these three experiments when different criteria were used to identify DEGs. These criteria included genes with a false discovery rate (FDR) of less than 0.05 (**B**), genes that were upregulated by 500% or more (**C**), genes that were upregulated by 200% or more (**D**), genes that were upregulated by 150% or more (**E**), genes that were increased by at least three standard deviations (**F**), genes that were increased by at least two standard deviations (**G**), genes that were increased by 150% or more and by more than three standard deviations (**H**), and genes that were increased by 150% or more and by more than two standard deviations (**I**). In each case, there was a highly significant overlap but also a number of non-overlapping DEGs. Percentage overlap indicates the percentage of overlapping DEGs divided by the number of genes in the smallest gene set. The number of overlapping genes is indicated by white text, while the number of genes unique to gene sets is indicated in black text.

**Figure 2 biology-12-00812-f002:**
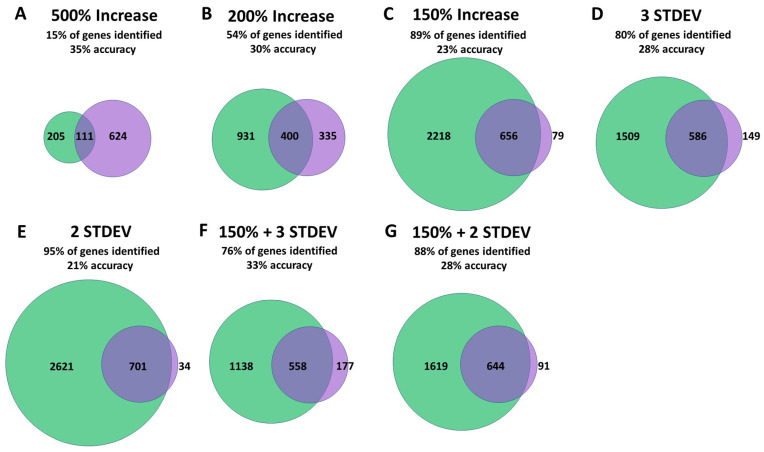
RNA sequencing of pooled samples is effective at identifying differentially expressed genes. To determine the ability of sequencing pooled RNA samples to identify differentially expressed genes (DEGs), we compared genes identified by pooled RNA-seq to genes found to be significantly upregulated in two experiments in which RNA samples were sequenced individually (false discovery rate (FDR) < 0.05). The criteria to identify upregulated genes in the pooled RNA-seq experiment included: genes that were upregulated by 500% or more (**A**), genes that were upregulated by 200% or more (**B**), genes that were upregulated by 150% or more (**C**), genes that were increased by more than three standard deviations (**D**), genes that were increased by more than two standard deviations (**E**), genes that were increased by 150% or more and by more than three standard deviations (**F**), and genes that were increased by 150% or more and by more than two standard deviations (**G**). The percentage of genes identified indicates the percent of genes identified in the pooled RNAseq sample divided by the number of genes that were found to be significantly upregulated in the two experiments in which samples were sequenced individually (735 genes total). The percentage accuracy was calculated by dividing the number of genes identified that are among the 735 genes that were significantly upregulated in the two experiments with individually sequenced RNA samples by the total number of genes identified. The criteria used to identify upregulated genes in the pooled RNA-seq experiment are indicated above the corresponding Venn diagrams.

**Figure 3 biology-12-00812-f003:**
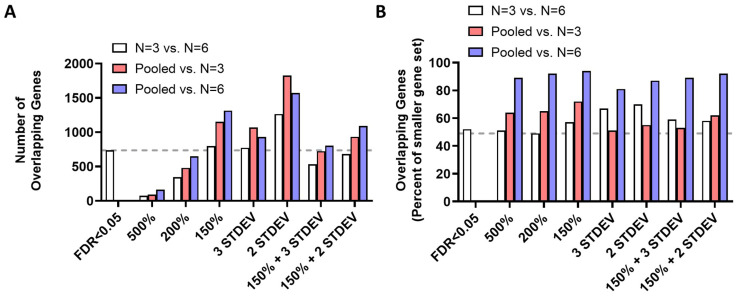
Overlap in upregulated genes identified between pooled RNA-seq and individually sequenced RNA-seq experiments is similar to overlap between two individually sequenced RNA-seq experiments. To determine the efficacy of identifying differentially expressed genes (DEG) by sequencing pooled RNA samples, we compared the total number of overlapping DEGs (**A**) and the percent overlap of DEGs (**B**) between RNA samples that were sequenced individually with 3 or 6 biological replicates and RNA samples in which 6 biological replicates were pooled together. As a benchmark for comparison, we examined the overlapping DEGs that were identified using a false discovery rate (FDR) of 0.05 in the individually sequenced RNA-seq experiments. Multiple criteria, including genes upregulated by 150%, genes upregulated by 2 standard deviations (STDEV), and genes upregulated by 3 STDEV, resulted in at least as many overlapping genes as the FDR < 0.05 criteria. For all of the different criteria examined, the number of overlapping genes with the pooled RNA sample was greater than between the two individually sequenced samples. The percentage overlap achieved using the FDR < 0.05 criteria was equalled or bettered by all of the other criteria. For all of the criteria examined, the greatest percentage overlap occurred between the pooled RNA sample and the individually sequenced RNA sample with six biological replicates. Overall, examining the overlap between the pooled RNA sample and the individually sequenced RNA sample was at least as effective as examining the overlap between two individually sequenced RNA samples.

**Figure 4 biology-12-00812-f004:**
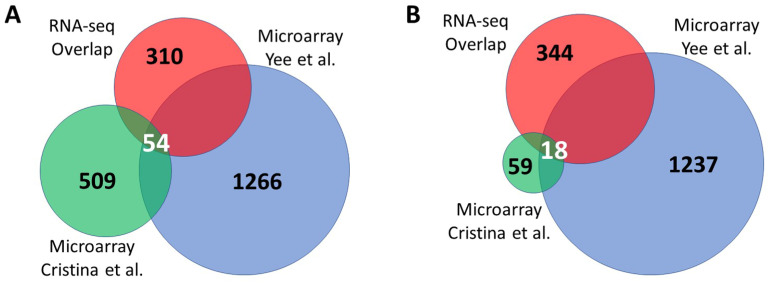
Comparison between RNA sequencing results and microarray. Genes that were identified as significantly upregulated or downregulated in previous microarray studies were compared to differentially expressed genes identified by RNA sequencing studies. For the RNA sequencing gene set, the overlap between two RNA sequencing experiments was used. Overall, there was only a small number of genes that were found to be significantly upregulated (**A**) or downregulated (**B**) in all four studies. The number of overlapping genes is indicated by white text, while the number of genes unique to gene sets is indicated in black text [25,26].

## Data Availability

Raw RNA sequencing data is available on NCBI GEO accession numbers GSE95240 (https://www.ncbi.nlm.nih.gov/geo/query/acc.cgi?acc=GSE95240, accessed on 13 May 2023) and GSE93724 (https://www.ncbi.nlm.nih.gov/geo/query/acc.cgi?acc=GSE93724, accessed on 13 May 2023).

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
