# Peer review of "RNA Sequencing of Pooled Samples Effectively Identifies Differentially Expressed Genes"

_biology, 2023, doi:10.3390/biology12060812_

Round 1
Reviewer 1 Report
In this study, authors examined the efficacy of identifying differentially expressed genes by sequencing a pooled RNA sample from long-lived isp-1 mitochondrial mutant worms, demonstrating that RNA sequencing of pooled RNA samples can be used to identify differentially expressed genes. As a communication, this paper is well-organized and well-written. However, I have some concerns about the innovative and scientific significance of this paper.
1. To my knowledge, “pooled RNA samples” is not rare and are a common strategy in some commercial sequencing companies. In fact, for some very small animals, companies often take the initiative to perform "pooled RNA samples" strategy. The reason is to circumvent individual differences. So, I hope the authors can better clarify the scientific value of this paper in the revised version. This is critically important for this paper
2. The authors use only one animal, the worm, is this conclusion convincing enough?
3. Again, I suggest that the authors better discuss the innovativeness of this paper, which I think has no problems with the writing and data, but does not better show the innovation and scientific value.
Author Response
Thank you for your comments and suggestions for improvement. Please see below for a point-by-point response.
1. To my knowledge, “pooled RNA samples” is not rare and are a common strategy in some commercial sequencing companies. In fact, for some very small animals, companies often take the initiative to perform "pooled RNA samples" strategy. The reason is to circumvent individual differences. So, I hope the authors can better clarify the scientific value of this paper in the revised version. This is critically important for this paper.
When performing RNA sequencing in C. elegans, tens to thousands of individual worms are collected to isolate mRNA for each individual replicate. This is considered to be one individual sample. Typically, three or more of these individual samples, each collected from tens to thousands of individual animals, are sequenced individually to identify genes that are significantly differentially expressed between different genotypes or interventions. In this manuscript, we have compared this approach with pooling the samples together prior to sequencing. Our results suggest that pooling prior to RNA sequencing still allows investigators to identify differentially expressed genes. We have updated the manuscript to clarify this point. This is also illustrated more clearly in the workflow diagram in Figure S1.
2. The authors use only one animal, the worm, is this conclusion convincing enough?
We apologize for the confusion. Each individual sample contains hundreds to thousands of individual worms. These individual samples were then either sequenced individually or pooled prior to sequencing. We have clarified this point in the revised manuscript.
3. Again, I suggest that the authors better discuss the innovativeness of this paper, which I think has no problems with the writing and data, but does not better show the innovation and scientific value.
In our review of the literature, we did not come across a paper that specifically compares sequencing pooled RNA samples versus sequencing RNA samples individually. In addition, our comparison between multiple experiments examining differentially expressed genes in long-lived isp-1 mutants has allowed us to generate a high confidence list of genes that are differentially expressed in these worms. We believe, this will be of interest to researchers studying the biology of aging.
Reviewer 2 Report
Dear authors,
Your communication, "RNA sequencing of pooled samples effectively identifies differentially expressed genes", shows the results of pooled and individual analysis of long-lived C. elegans mutants. After using previously published data, you determine that by analyzing a pooled approach, researchers can obtain similar relevant results reducing their time and costs. I find this manuscript relevant to their field. However, I would like to comment on some concerns:
Major comments
1. Please, describe the origin of all worms used in this study.
2. I suggest adding quality values and concentration of RNA after their extraction.
3. I was not able to see any figures. Could you follow the authors' guidelines for attaching them, please?
4. Please, add a workflow for facilitating the understanding of your findings.
5. I suggest adding a discussion about the potential limitations of pooled samples. It seems to work in culture-based samples, but would it be the same in individual/patient samples?
Minor comments
6. Please, check the descriptions for some abbreviations. For example, "qPCR" stands for quantitative PCR, whereas "qRT-PCR" stands for quantitative reverse-transcriptase PCR (lines 58-59)
Author Response
Thank you for your comments and suggestions for improvement. Please see below for a point-by-point response.
1. Please, describe the origin of all worms used in this study.
We have updated the manuscript to describe the origin of the worms used in this study. We used N2 wild-type worms, which were isolated in Bristol and commonly used by C. elegans researchers. We also used long-lived isp-1 mutants, which were generated in a screen for slow development performed in Siegfried Hekimi’s lab (Feng et al., 2001).
2. I suggest adding quality values and concentration of RNA after their extraction.
RNA quality and concentration was initially assessed using a Nanodrop spectrophotometer in our laboratory and then an Agilent Bioanalyzer in the Genomics Core at the Van Andel Research Institute. We have added these details to the revised manuscript. Unfortunately, as this study is comparing datasets generated in the past at a previous Institution, we no longer have the details about the RNA quality and concentration.
3. I was not able to see any figures. Could you follow the authors' guidelines for attaching them, please?
We apologize for this. Initially the figures were uploaded separately from the manuscript text. The figures have now been integrated into the text.
4. Please, add a workflow for facilitating the understanding of your findings.
We have added a workflow diagram as Figure S1 to make it easier for readers to understand our findings.
5. I suggest adding a discussion about the potential limitations of pooled samples. It seems to work in culture-based samples, but would it be the same in individual/patient samples?
This is an excellent point. In our study we are using a genetic model organism in which all individuals are genetically identical. In applications using human samples or samples derived from animals in the wild, each sample has a unique genome that is contributing to their gene expression and pooling samples may result in the loss of information about variability caused by differences between genomes.
Minor comments
6. Please, check the descriptions for some abbreviations. For example, "qPCR" stands for quantitative PCR, whereas "qRT-PCR" stands for quantitative reverse-transcriptase PCR (lines 58-59)
We have modified the manuscript to use qRT-PCR throughout.
Reviewer 3 Report
In this manuscript, the authors proposed an economic way of RNA-seq using the pooled RNA sample. The authors compared the overlap of differentially expressed genes (DEGs) between the pooled RNA sample and individually sequenced RNA samples. It showed a high overlap rate between pooled and individually sequenced RNA samples, indicating a high efficacy in the identification of DEGs using pooled RNA samples. Overall the findings from this manuscript provide beneficial information to the sequencing field. Meanwhile, there are several concerns that might weaken the manuscript:
1. Experiment design. The authors only sequenced one group (ips-isoform) with pooled RNA sample, which make it not realistic. The more reasonable design would be to pooled six biological replicates of RNA samples into one RNA sample for both groups. Then identify the DEGs based on the two pooled RNA samples and compare the overlap of DEGs identified from pooled RNA samples and individually sequenced RNA samples.
2. In the Methods section, the authors should describe in more details how the differentially expressed genes (DEGs) were identified, provide more details on the parameters, such as the cutoff of fold change and adjusted p-value. Especially, in the case of one replicate (pooled RNA samples), how to identify DEGs?
In line 137-138, "In some cases, genes that were previously reported to be differentially expressed are no longer considered to be genes." I'm confused about the exact meaning of this sentence. Please clarify.
Author Response
Thank you for your comments and suggestions for improvement. Please see below for a point-by-point response.
1. Experiment design. The authors only sequenced one group (isp-isoform) with pooled RNA sample, which make it not realistic. The more reasonable design would be to pooled six biological replicates of RNA samples into one RNA sample for both groups. Then identify the DEGs based on the two pooled RNA samples and compare the overlap of DEGs identified from pooled RNA samples and individually sequenced RNA samples.
In designing this experiment, we wanted to determine whether using percent change would be sufficient to identify differentially expressed genes, or whether it would be more accurate to take variability into account by using standard deviation. In order to determine standard deviation, we needed to individually sequence the wild-type samples. Our results suggest that it should be possible to just use percentage change to identify differentially expressed genes in which case it should be possible to sequence pooled samples for wild-type as well. We have added these points to the revised manuscript.
2. In the Methods section, the authors should describe in more detail how the differentially expressed genes (DEGs) were identified, provide more details on the parameters, such as the cutoff of fold change and adjusted p-value. Especially, in the case of one replicate (pooled RNA samples), how to identify DEGs?
According to this suggestion, we have expanded the description of how we identified differentially expressed genes in the methods section.
3. In line 137-138, "In some cases, genes that were previously reported to be differentially expressed are no longer considered to be genes." I'm confused about the exact meaning of this sentence. Please clarify.
This statement indicates that when the microarray studies were performed, they reported the differential expression of a gene, which is now no longer considered to be a gene. In some cases, this was because the putative gene originated from a transposon and thus was not a C. elegans genes. In other cases, it was later decided that there was not enough evidence that it was a gene, so it was declared a dead gene. We have clarified this in the revised manuscript.
Round 2
Reviewer 1 Report
The authors have answered my doubts well, and I believe that this study does have some innovative and scientific value. I think this paper is currently ready for publication in Biology.
Reviewer 2 Report
Dear authors,
Your communication, "RNA sequencing of pooled samples effectively identifies differentially expressed genes", shows the results of pooled and individual analysis of long-lived C. elegans mutants. After using previously published data, you determine that by analyzing a pooled approach, researchers can obtain similar relevant results reducing their time and costs. I find this manuscript relevant to their field. Thank you for having addressed my previous comments.